# Assessing the Safety of Mechanically Fibrillated Cellulose Nanofibers (fib-CNF) via Toxicity Tests on Mice: Single Intratracheal Administration and 28 Days’ Oral Intake

**DOI:** 10.3390/toxics12020121

**Published:** 2024-02-01

**Authors:** Yoshihiro Yamashita, Akinori Tokunaga, Koji Aoki, Tamotsu Ishizuka, Hideyuki Uematsu, Hiroaki Sakamoto, Satoshi Fujita, Shuichi Tanoue

**Affiliations:** 1Research Center for Fibers and Materials, University of Fukui, 3-9-1 Bunkyo, Fukui 910-8507, Japan; tanoue@u-fukui.ac.jp; 2Life Science Research Laboratory, University of Fukui, 23-3, Matsuoka Shimoaizuki, Eiheiji-cho, Fukui 910-1193, Japan; aktoku@u-fukui.ac.jp; 3Organization for Life Science Advancement Programs, University of Fukui, 3-9-1 Bunkyo, Fukui 910-8507, Japan; aokik@u-fukui.ac.jp (K.A.); tamotsui@u-fukui.ac.jp (T.I.); uematsu@matse.u-fukui.ac.jp (H.U.); hi-saka@u-fukui.ac.jp (H.S.); fujitas@u-fukui.ac.jp (S.F.); 4Department of Pharmacology, Faculty of Medicine, University of Fukui, 23-3, Matsuoka Shimoaizuki, Eiheiji-cho, Fukui 910-1193, Japan; 5Third Department of Internal Medicine, Faculty of Medicine, University of Fukui, 23-3, Matsuoka Shimoaizuki, Eiheiji-cho, Fukui 910-1193, Japan; 6Department of Frontier Fiber Technology and Science, Graduate School of Engineering, University of Fukui, 3-9-1 Bunkyo, Fukui 910-8507, Japan

**Keywords:** cellulose, nanofiber, in vivo, TG412, TG407, safety

## Abstract

Mechanically fibrillated cellulose nanofibers, known as fib-CNF (fiber length: 500 nm; diameter: 45 nm), are used in composites and as a natural thickener in foods. To evaluate their safety, we conducted a 28-day study in mice with inhalation exposure at 0.2 mg/body and oral administration of 400 mg/kg/day. Inhalation exposure to fib-CNF caused transient weight loss, changes in blood cell counts, and increased lung weights. These changes were attributed to adaptive responses. The oral administration of fib-CNF for 28 days resulted in no apparent toxic effects except for a slight decrease in platelet counts. The fib-CNF administration using the protocols studied appears to be safe in mice.

## 1. Introduction

Mechanically fibrillated cellulose nanofibers (fib-CNF) are widely used as a substitute for plastics and glass fibers, and as a natural food additive to improve thickening properties. Pinto et al. [1] reported that cellulose microfibers (CMF), nanofibers (CNF) and nanocrystals (CNC) exhibited higher stability and lower cytotoxicity compared to those of multi-walled carbon nanotubes (MWCNTs). Li et al. [2] reported that CNF had a low cellular uptake rate and low cytotoxicity, whereas CNC were readily taken up by cells and exhibited cytotoxic effects. We hypothesized that fib-CNF was not readily taken up by cells.

Fujita et al. [3] recently evaluated the safety of CNF in rats by intratracheally administering different forms of CNF at 2.0 mg/kg and assessing their effects on lung inflammation over 90 days. Their results showed that administration of fib-CNF, compared to that of TEMPO-oxidized or phosphate-esterified CNF, resulted in a milder acute alveolar inflammatory response that was further suppressed after 90 days. De Lima et al. [4] found that TEMPO-oxidized CNF showed minimal cytotoxic effects, suggesting that these fibers have a promising safety profile for health and medical applications. Shazali et al. [5] showed that CNC have low cytotoxicity against rat C6 glioma cells and normal NIH3T3 fibroblasts. Salari et al. [6] also reported that CNF and FITC-CNC, which contain fluorescein isothiocyanate (FITC) molecules, did not exhibit cytotoxicity. However, Ventura et al. [7] showed immunotoxic and genotoxic effects of TEMPO-oxidized CNF.

Ong et al. [8] fed fib-CNF mixed with food to rats for 90 days in accordance with TG408. They reported that no adverse effects were observed: the fiber diameter of CNF was 25.06 ± 6.29 nm and the fiber length was 227.7 ± 103.3 nm. However, CNF was mixed into the feed as a dry powder and hardened into pellets, which are believed to be an aggregate rather than being in the native needle-like or fibrous structure of CNF.

Nagai et al. [9] argued, in terms of nano-sized durable fibrous materials, that there is a lack of data on the interaction between the macrophage and mesothelium, especially in vivo. This may be due to the inherently heterogeneous properties of nanofibrous materials. Variations in nanofibers include length, diameter, surface area, density, shape, the presence of mixed metals (including iron), and collagen content.

Since 2009, the European Cosmetics Regulation has prohibited animal testing for cosmetic ingredients and the marketing of products tested on animals. As a result, in vitro toxicity assessment and cell testing have become the primary alternatives to animal testing. However, it is important to note that the lack of comprehensive in vivo data, especially given the potential diverse applications of fibrous nanomaterials in various fields, underscores the need for new standardized testing methods. Such methods are essential to minimizing risks to human health, especially since two of the main mechanisms of concern—macrophage activation and mesothelial cell damage—are both associated with inflammation and cancer.

In Japan, animal testing of CNF for cosmetic applications has been phased out due to animal welfare concerns. However, there is still a need to evaluate the safety of CNF when they are used to reinforce composite materials, to replace plastics, or as food additives. Initially, we shared the view that it would be preferable to assess the safety of fib-CNF for human health using cell experiments. However, we reconsidered the importance of in vivo testing using animal models due to conflicting reports on cytotoxicity as well as the results of the rat study by Fujita et al. [3], which indicated the potential of inhalation toxicity of CNF.

CNF is used as a reinforcing material in composite materials as a substitute for glass fiber, and its thixotropic properties and dispersion stability make it effective in preventing caking and dripping when mixed with paints and coatings. Aerosols and dusts of CNF are generated in the process of manufacturing or disposing of CNF. When added to products as a thickening agent, they can also be inhaled by humans. Inhalation testing in the natural state through the nose or mouth, such as whole-body exposure or nasal exposure, is the most desirable method, but it is not easy to quantify whether CNF has entered the lungs with these testing methods. An alternative method is direct administration into the lungs. Intratracheal administration is the simplest and most widely used method to ensure the quantitative delivery of nanofibers into the lungs [10]. This technique is applicable not only to nanomaterials but also to airborne and potentially inhaled substances, if they can be aerosolized [11,12,13,14,15]. Ede et al. [16] presented a review of recent animal studies on the intratracheal administration of CNF and CNC.

In this study, we used mice to evaluate the safety of fib-CNF, particularly in its role as a plastic replacement and food additive for applications beyond cosmetics. In an in-tratracheal administration experiment, we administered a single dose of fib-CNF intratracheally to 10-week-old male C57BL/6JJmsSlc mice and performed necropsy 3 and 28 days after administration to assess toxicity. In a separate oral study, 400 mg/kg/day of fib-CNF was gavaged into the stomach of mice for 28 consecutive days.

## 2. Experiments

### 2.1. Materials

#### fib-CNF

Mechanically fibrillated CNF extracted from wood cellulose pulp via sterilization at 80 °C was used (fib-CNF, BiNfi-s FMa-10002, Lot ML01-L02, Sugino Machine, Toyama, Japan). The potential impurities in fib-CNF (BiNFi-s) are as follows: general bacteria < 10 colony-forming units/mg, *E. coli* negative, ash < 0.1 wt% (Hygienic Testing Method commentary 2015 Japan)), heavy metals < 0.01 mg/kg (ICP-MS), and arsenic < 0.01 mg/kg (ICP-MS). The same stock fib-CNF was used for both single intratracheal and oral forced intragastric administration.

To improve the stability of the fib-CNF aqueous solution [17], a fib-CNF dispersion solution that did not sediment even after standing for 1 week was obtained via high-speed stirring at 8000 rpm for 20 min in a homogenizer (HG-200; AS ONE Corporation, Osaka, Japan).

The fiber length and fiber width of fib-CNF were measured using a Hitachi H-7650 No. 2 transmission electron microscope and a JEOL JSM-7600F field emission scanning electron microscope, respectively, as well as Image-J (ImageJ-win64) analysis software.

### 2.2. Single Intratracheal Instillation

#### 2.2.1. Mice

Thirty-two C57BL/6JJmsSlc (SPF animals) mice at the age of 9 weeks were obtained from Japan SLC Co. (Shizuoka, Japan) and kept in the animal facility of the Univ. of Fukui for 1 week before the experiments were performed. Among them, we used 20 mice whose weight range at the beginning of the experiment (at the age of 10 weeks) was within ±20% of the mean body weight and who appeared healthy.

Solid-feed MF (Oriental Yeast Industry Co., Ltd., Tokyo, Japan) was placed in a feeder attached to the cage lid and consumed by mice ad libitum. Drinking water was provided from tap water using a clear water bottle. The group of mice, test substance, dosage, concentration of administered solution, dosage volume, number of mice per group, and date of necropsy are shown in Table 1.

Fujita et al. set the limit concentration of fib-CNF for intratracheal administration in rats at 5 mg/kg (227~307 g body weight) [3]. Few deaths occurred at this concentration. Therefore, for our present experiments we used 0.2 mg/body (7 mg/kg) as the maximum concentration in mice.

#### 2.2.2. Route of Administration

A single intratracheal instillation was chosen to administer all the test substance into the lungs.

The amount of test substance (2 wt% deionized water solution) was measured using a measuring cylinder and then added to the prescribed volume with distilled water (lot no. 1G70N; Otsuka Pharmaceutical Co., Ltd., Tokyo, Japan) to obtain the 100 μL of 0.2 wt% distilled water solution as shown in Table 1.

On the day before the start of the experiment, the animals were assigned to groups and a single intratracheal dose was administered under isoflurane (Mylan Inc., Tokyo, Japan) anesthesia using NARCOBIT-E (KN-1071; Natsume Seisakusho, Tokyo, Japan), a simple inhalation anesthetic device for small laboratory animals. Administration was performed using a disposable syringe connected to a DIMS-type endotracheal sonde for mice. Necropsy was performed 3 or 28 days after administration.

#### 2.2.3. Observation

##### General Condition

Beginning on the first day of the observation period, all animals were observed twice a day (post-dose or in the morning and afternoon) to assess their general condition and intoxication symptoms. These observations were recorded for each individual animal.

##### Body Weight

All animals were individually weighed using a Model LA4200 electronic balance (Sartorius Co., Ltd., Tokyo, Japan) immediately before (day 0) and 3, 7, 14, 21, and 28 days after administration. Body weights were also measured at the time of sacrifice.

#### 2.2.4. Blood Tests (Table 2)

The following tests were performed on all surviving animals at the end of the test substance administration period. Blood samples were collected from the abdominal aorta after dissection of the abdomen under isoflurane anesthesia using NARCOBIT. The order of sacrifice and blood collection of animals was determined by animal identification number, from smallest to largest, as follows:

01M01→02M01→03M01→04M01→01M02→02M02→03M02→04M02→…

**Table 2 toxics-12-00121-t002:** Blood toxicity tests.

Item		Measurement Method
(1)	Red blood cell count (RBC)	Sheath flow DC detection method
(2)	Hemoglobin (HGB)	SLS hemoglobin method
(3)	Hematocrit value (HCT)	Sheath flow DC detection method
(4)	Mean corpuscular volume (MCV)	Sheath flow DC detection and calculation method
(5)	Mean corpuscular hemoglobin (MCH)	Sheath flow DC detection and calculation method
(6)	Mean corpuscular hemoglobin concentration (MCHC)	Sheath flow DC detection and calculation method
(7)	Platelet count (PLT)	Sheath flow DC detection method
(8)	Reticulocyte count (RBC)	Flow cytometry method and calculation method
(9)	White blood cell count (WBC)	Flow cytometry method
(10)	White blood cell count by type	
	Lymphocyte count	Flow cytometry method
	Neutrophil count	Flow cytometry method
	Eosinophil count	Flow cytometry method
	Basophil count	Flow cytometry method
	Monocyte count	Flow cytometry method

A portion of the collected blood was transferred to a test tube containing EDTA-2K anticoagulant and analyzed with a multiparameter automated blood cell analyzer, XT-2000i (Sysmex Corporation). The sheath flow DC detection method detects changes in the current passing through the sheath fluid, and flow cytometry analyzes cell characteristics or properties using laser irradiation. The SLS hemoglobin detection method uses cyanide-free sodium lauryl sulfate (SLS) to lyse red and white blood cells in the specimen, altering the globin and oxidizing the heme group.

#### 2.2.5. Pathological Examinations

The following pathological examinations were performed on animals that underwent blood sampling and then were bled to death at the end of the 3 or 28 days’ post-dose observation period.

##### Partial Examination

The lungs (including bronchi), trachea, and any organs or tissues with abnormal gross findings were removed and fixed in 10% buffered formalin solution.

##### Organ Weights

The lungs (including bronchi) of all surviving animals were weighed using an electronic balance, Model CP323S (Sartorius Co., Ltd. Goettingen, Germany), and the organ weight-to-weight ratio was calculated using the necropsy-day body weight.

##### Histopathological Examination

For all animals, each of the left and right lobes (upper, middle, lower, and accessory lobes) of the lungs (including bronchi) was cut from the largest surface, paraffin-embedded, and thinly sectioned in accordance with the usual method. The specimen was used for the histopathological examination of the lungs, stained with hematoxylin-eosin.

The histopathology of single intratracheally administered lungs and orally administered stomachs was performed by the expert Dr. Akihiro Hagiwara (DIMS Medical Research Institute, Inc. 64, Nishi-Asai-go-ura, Asai-cho, Ichinomiya-shi, Aichi 491-0113, Japan).

Macrophage aggregation in pathological tissues in the observed images was evaluated on a 5-point scale as follows: None: no difference from the control group; Minimal: one cluster is seen in one location; Minor: a few clusters are seen; Moderate: widespread clusters; Severe: numerous clusters throughout.

#### 2.2.6. Statistical Processing

Statistical significance tests between the control and test substance-treated groups (groups 01M, 02M, 03M, and 04M) were performed on each autopsy date and determined at the 5% (*p* < 0.05) or 1% (*p* < 0.01) risk rate level. Statistical analysis was performed on the body weight, hematology, and organ weight of the control and test substance-treated groups using the F test, Student’s *t*-test (two-tailed) in the case of equal variance, and Welch’s test (two-tailed) in the case of unequal variance. Fisher’s (one-tailed) direct probability test was used to test for differences in the frequency of occurrence at autopsy and at histopathological examination.

### 2.3. Oral Administration

#### 2.3.1. Mice

Twenty-three male C57BL/6JJmsSlc mice (SPF animals) were purchased from Japan SLC Co. The mice were purchased at 7 weeks of age, and experiments were conducted starting at 8 weeks of age. The mice were fed solid-feed MF (Oriental Yeast Industry Co.) in a feeder attached to the cage lid ad libitum. Drinking water was provided ad libitum from tap water using a clear water bottle.

Regarding oral intake, Nagano et al. [18] reported that 15.2 ± 0.9 and 21.3 ± 0.8 mg/mice body/day of CNF was administered to mice for 7 weeks with no observed health effects. However, in their experiment, the mice drank 9 to 11 mL of water per day, and the amount of CNF dispersed in that water was the estimated intake at 0.1 wt%. However, while this method is simple, it is also inaccurate due to the inevitable spillage during drinking.

The OEDC Guideline for “repeated dose 28-day oral toxicity study in rodents” (TG407) sets a threshold dose of 1000 mg/kg/day (25 mg/mice body/day). Generally, 300 mg/kg is the threshold dose for toxic substances in the criteria for toxic substances. For our experiments, we chose a dose of 400 mg/kg/day of fib-CNF (equivalent to 10 mg/mice body/day), since it was required that the fib-CNF be administered directly into the stomach without vomiting during a single forced oral administration using a sonde.

The groups of mice, test substance, dosage, concentration of administered solution, number of mice per group are shown in Table 3. The concentration of fib-CNF in the solution diluted with ion exchanged water is 2 wt%. The forced oral dose for each dose is 20 mL/kg/day. The solid content of fib-CNF in 20 mL of aqueous solution is 400 mg/kg/day. In comparison, humans eat 20~25 g of dietary fiber per day. Among these, insoluble dietary fiber, which is insoluble in water, is abundant in foods such as soybeans, burdocks, and grains. It is said that about 80% of the dietary fiber consumed by Japanese people is insoluble fiber, most of which is cellulose. If the average Japanese adult weighs 50 kg, this means that they consume 400 mg/kg/day of cellulose [19].

The test substance dosage was determined as the maximum dose that could be administered, following the previously reported method The route of administration was oral, since humans ingest the test substance orally, and the forced oral administration method was chosen to ensure that the prescribed dose could be administered.

#### 2.3.2. Histopathological Examination

Organs and tissues were collected from all control and test substance-treated groups (groups 01N and 02N), including both deceased and ailing animals. These specimens were then paraffin-embedded, thinly sliced, and subjected to hematoxylin and eosin staining. Subsequently, we conducted examinations using a speculum, following standard procedures.

## 3. Results

### 3.1. Morphological Observation of fib-CNF

Figure 1a is an image of fib-CNF observed via TEM, and Figure 1c shows the results of the measurement of fiber length distribution based on these five images in different regions. Figure 1b is an image of fib-CNF observed via FE-SEM, and similarly, Figure 1d shows the measurement results of fiber width distribution. fib-CNF appear needle-like from the TEM image, while they resemble noodles in the FE-SEM image. The mean fiber length was 620 nm (standard deviation (S.D.): 368 nm), while the average fiber diameter was 42.9 nm (S.D.: 9.2 nm). This difference was due to the different regions visualized via the imaging modalities, with the TEM images allowing us to observe the interior of the sample, and the SEM images permitting a visualization of the surface of the sample in three dimensions. The fib-CNF used by Fujita et al. was produced by Daio Paper Corporation (Tokyo, Japan) with a fiber length of 1700 nm and a fiber diameter of 21.2 nm [3]. Compared to this, the fib-CNF from the Sugino Machine Corporation used here are of a shorter fiber length grade. On the other hand, the phosphorylated CNF (Oji Holdings Corporation, Tokyo, Japan) and TEMPO-oxidized CNF (Nippon Paper Industries, Tokyo, Japan) used by Fujita et al. have an average fiber length of 800~1000 nm and fiber diameter of 7.0~8.0 nm [3]. These are close to the fiber lengths of the fib-CNF used here.

### 3.2. Toxicity of a Single Intratracheal Instillation of fib-CNF to Mice after 3 and 28 Days

#### 3.2.1. Survival and General Condition

No deaths were observed in any of the groups from the administration of the test substance through the observation period, and the survival rate at the end of the 3- or 28-day post-dose observation period was 100% in all groups. Regarding the general condition of the animals, a “wheezing” or “hissing” sound caused by intratracheal administration was observed in all animals just after administration, and only one animal in the test substance group made this sound by the afternoon observation on the day of administration. In addition, all animals in the test substance group showed a decrease in spontaneous locomotion in the afternoon of the day of administration, but no abnormalities, including a wheezing/hissing sound, were observed on the following day. Therefore, all changes were transient and caused by intratracheal administration.

#### 3.2.2. Body Weight

Individual values for each group are shown in Figure 2. The test substance group showed significantly lower values than those of the control group after 3 days of treatment, but recovered after 7 days of treatment. There was a significant difference between the control and the CNF intratracheal groups, with a *p*-value of 0.03 in the ANOVA. This was due to a rapid decrease in body weight after 3 days of administration.

#### 3.2.3. Hematological Examination

The mean value and standard deviation for each group are shown in Table 4 and Table 5. After 3 days of treatment, the test substance group showed a significantly higher eosinophil count and a significantly lower reticulocyte count compared to that of the control group. Both changes were transient, and no changes were observed after 28 days of treatment. A histopathologic examination of the lungs of the test group revealed that the statistically significantly higher eosinophil counts were not accompanied by increased eosinophil infiltration, and the lower reticulocyte counts were not accompanied by changes in the red blood cell count, hemoglobin, hematocrit, or erythrocyte constant count. Therefore, the cause of any of the changes is unknown, but since they were transient and relatively mild, they were of low toxicological significance. No statistically significant differences were found between the control and test substance-treated groups in other laboratory parameters.

#### 3.2.4. Pathological Examination

##### Necropsy Findings

No significant changes were observed in any of the animals at necropsy at 3 or 28 days post-dose.

##### Organ Weights

The mean and standard deviation for each group is shown in Table 6 and Table 7. At 3 and 28 days post-dose, we found significantly higher absolute and relative lung weights in the test substance groups in both cases.

##### Histopathological Examination

Histopathological findings for each group are shown in Table 8 and Table 9. The amount of macrophage aggregates on individual animals was calculated based on the relative evaluation of histopathological findings and assigned as follows: none, not remarkable, slight, moderate, and marked. At 3 days and 28 days post-treatment, all animals in the test substance treatment groups showed moderate aggregation of macrophages in the alveoli. The absence of inflammatory changes at either time point was also attributed to adaptive changes in response to the intratracheal administration of the test substance.

Figure 3 and Figure 4 show the histopathology of the lungs. CNF aggregates with a length of 5~30 μm were observed near the alveoli around the terminal bronchiole at 3 and 28 days after a single intratracheal administration of CNF, in addition to the aggregation of macrophages caused by CNF. These aggregates are clearly fewer in number and smaller in size at 28 days than at 3 days. They are simply a collection of CNF and are different from the aggregates of enlarged CNF phagocytosed by macrophages. The fib-CNF aggregates shown in the Figure 3 and Figure 4 also confirmed that CNF aggregates are easily distinguished by the absence of cell nuclei.

### 3.3. Toxicity of fib-CNF in Mice after 28 Days of Forced Oral Administration

#### 3.3.1. Survival and General Condition

One case (animal no. 01N04) in the control group was euthanized because it showed emaciation around day 12 of administration. At necropsy, the animal weighed 16.4 g, which was 32.5% less than the average weight of the control group (24.3 g) at 7 days post-dose. At necropsy, gross examination revealed black contents in the stomach and small intestine. Consistently, histopathological examination indicated the presence of moderate erosions in the glandular stomach. However, as there were no discernible signs suggesting an administration error, the mechanism of causing the change in the animals’ general condition was unclear. No abnormal findings in the general condition of the other animals were observed throughout the administration period.

#### 3.3.2. Body Weight

The mean body weight and standard deviation for each group are shown in Figure 5. Both the control and test groups showed a steady trend, and there was no statistically significant difference between the two groups.

#### 3.3.3. Hematological Examination

The hematological examination results for each group are shown in Table 10. Significantly lower platelet counts (PLT) were observed in the test group compared to those in the control group. No other hematological findings suggestive of bleeding or a bleeding tendency were observed in the test group. The degree of hemorrhage was very slight and within the range of variability, so it was considered unlikely to have been caused by the administration of the test substance. No statistically significant differences were found between the control and test substance groups for the other items.

#### 3.3.4. Necropsy

The dead animal (animal no. 01N04) had black content in the stomach and small intestine. No abnormal findings were observed at necropsy in the surviving animals, including those treated with the test substance.

#### 3.3.5. Histopathological Examination

The histopathological findings for each group show that moderate erosions were observed only in the gastric mucosa of the animal that died in the control group (animal no. 01N04). There were no other changes in the gastrointestinal tract in another animals. Additional information can be found in Appendix A.

Figure 6 shows the glandular stomach tissue of the control and CNF-treated mice; 28 days of oral administration of CNF did not cause any abnormalities in the tissues of the mice.

## 4. Discussion

### 4.1. Single Intratracheal Instillation

In this study, fibrous cellulose nanofibers (fib-CNF) were administered intratracheally to male C57BL/6JJmsSlc mice at a dose of 0.2 mg per body weight (near the maximum concentration limit). The purpose was to evaluate the potential toxicity associated with the intratracheal administration of the test substance. Throughout the observation period, no deaths occurred in either the control or test groups. In addition, there were no detectable changes in general health, body mass, or hematologic parameters suggestive of test substance toxicity.

Higher absolute and relative lung weights were observed in the experimental group at either 3 or 28 days after dosing. Histopathologic examination revealed a moderate presence of alveolar macrophages at these time points. These observed changes were considered to be adaptive responses induced by the intratracheal administration of fib-CNF.

Ilves et al. [20] reported that fib-CNF induced an innate immune response 24 h after intratracheal administration, causing characteristics of a Th2-type inflammation, with a modest immune response still present 28 days later.

Catalán et al. [21] reported that mice exposed to a single intratracheal dose of 0.2 mg/body of TEMPO-oxidized CNF showed increased numbers of neutrophils, macrophages, lymphocytes, and eosinophils. The histopathology of their alveoli was like that under our intratracheal administration of fib-CNF at 3 days (Figure 3d). They also reported that DNA damage was observed in isolated lung cells, but no dose-response relationship was observed; no DNA damage was observed in cells isolated from BAL, and no chromosomal damage was observed in bone marrow erythrocytes.

Macrophage accumulation is a common response to the inhalation of sufficiently large foreign particles capable of entering the alveoli [22,23,24,25]. However, the critical consideration is the presence of tissue thickening, which may have carcinogenic potential following macrophage accumulation. This aspect is considered crucial. The process of collagen fiber production and subsequent fibrosis was not observed during the observations made at 3 and 28 days after a single intratracheal administration of fib-CNF. Therefore, this study suggests that macrophage accumulation in the alveoli mirrors the phenomenon observed as a result of the inhalation of nano-sized foreign particles, bacteria, or viruses.

The evaluation of the inhalation toxicity of fib-CNFs should be performed as shown in the present study, and more detailed acute and chronic effects should be studied in accordance with OECD guidelines TG412 and TG413. In addition, BALF studies should be performed along with lung histopathologic observations. Additional validation is needed to assess the stiffness and shape persistence of CNF and to evaluate the immune and inflammatory responses in the lung when fib-CNF is present for prolonged periods.

### 4.2. Oral Administration

Nanofibers are widely used in cosmetics, pharmaceuticals, and foods. Among these, animal studies on the health effects of nanofibers taken orally as food are still scarce. Nanofibers are classified into different types, including nanowires, nanotubes, nanofibers, and nanohydrogels, based on their binding and structure. Surface functionalization further complicates their properties. The distribution of nanoparticles, including nanofibers, in the body varies depending on properties such as excretion and intracellular accumulation [26,27].

It is also important to note that fib-CNF and carboxymethylcellulose (CMC) are already widely used as food additives [28,29,30,31,32,33,34,35,36,37].

Oral administration toxicity tests are conducted according to the OECD TG407 guideline test [38,39,40,41,42,43,44]. In this study, we used mice instead of rats as a deviation from the TG407 guidelines. Rats typically weigh about 250 g, while mice weigh about 30 g, making them more cost-effective and practical to raise. Mice are the preferred choice of research institutions, including universities. The primary advantage of using small animals is that they require minimal resources, including food, water, and living space [45]. However, their anatomy differs significantly from that of humans. Larger animals have more human-like anatomy and physiology, but are more expensive and require more space. As a result, mice are often selected to perform 28-day oral toxicity studies [46,47,48,49,50,51,52,53].

In our study, fib-CNF were administered orally via gavage to male C57BL/6JJmsSlc mice at doses of 0 (vehicle control) and 400 mg/kg/day for 28 days. The objective was to evaluate the toxicity resulting from repeated oral administration.

Zhang et al. [54] performed toxicological studies on the oral uptake of different CMC in vivo. They administered a 1% or 3.5% w/w suspension of CMC via gavage to mice once daily for 8 weeks. The results showed no significant differences in hematological parameters or serum markers between the control group and the mice receiving CMC. However, CMC-fed mice exhibited a significant increase in fecal fat content associated with reduced food intake and body weight. These results suggest that CMC is non-toxic and has the potential to serve as a food additive or dietary supplement to reduce caloric intake.

Chen et al. [55] observed that CNF is used as a food additive or supplement in diets high in fat and sugar. However, long-term studies of the effects of CNF supplementation have revealed several notable findings. CNF was found to reduce intestinal fat absorption, attenuate the development of an induced fatty liver, to slightly reduce body weight, and to influence glucose homeostasis.

Zhang et al. used CNF made from cotton lint with a fiber length of about 1 μm and a fiber diameter of 20 nm. The CNF used by Chen et al. had a fiber length of several hundred nm and a fiber diameter of 50 nm, like the length and diameter of the fib-CNF used in this study.

No other abnormal general health findings were observed during the administration period in either the test substance group or the control group. In addition, no changes in body weight, necropsy findings, or histopathologic examinations were observed that would indicate the effects of test substance administration. In hematologic evaluations, a slight decrease in platelet count (PLT) was observed in the test substance administration group compared to that in the control group, but this decrease was of minimal magnitude and unlikely to be attributable to the test substance itself.

This study demonstrates that the repeated oral administration of fib-CNF at a daily dose of 400 mg/kg to male C57BL/6JJmsSlc mice for 28 days did not result in any discernible toxic changes attributable to the test substance.

The results show no biological effects of oral administration of CNF on male mice after 28 days. We could not confirm the weight loss effect observed by Zhang et al., and Chen et al. showed that the continuous administration of CNF to female mice slowed weight gain after 35 days [55]. They observed the weight loss effect in male mice after 4 weeks, but not in females. However, their dose was 0.75 mg/body weight/day, which was much lower than our dose of 10 mg/body weight (400 mg/kg). In summary, Zhang et al. showed that cotton-derived CNF with a fiber length of 1 μm had a weight loss effect in female mice, and Chen et al. showed that mechanically disintegrated CNF with a fiber length of less than 1 μm had a weight loss effect in male mice, but “no weight-loss effect” was observed in male mice in our study. Ong et al. [8] also reported, as we did, that NOAEL doses of 2194.2 mg/kg/day (males) and 2666.6 mg/kg/day (females) for 90 days did not result in weight loss compared with the control group.

The mice in the Chen et al. study were 8-week-old male C57BL/6 mice, and our mice were also the same 8-week-old male C57BL/6 mice. The difference between them may have been due to the different diets used for rearing. Although Chen et al. fed a high-glucose, high-fat Western diet (WD), which made the effects more apparent, we used a basal diet designed for general short-, medium-, and long-term rearing experiments. Future experiments should be conducted in accordance with the TG407 guidelines, including comparative experiments between males and females and experiments on the effects of feeding, to verify whether or not CNF is indeed safe and effective for weight loss.

## 5. Conclusions

In this study, we evaluated the biological safety of mechanically fibrillated cellulose nanofibers, referred to as fib-CNF, via inhalation and oral administration to mice over a 28-day period.

### 5.1. Inhalation and Intratracheal Administration of fib-CNF in Mice

The toxicity of fib-CNF was evaluated by administering a single intratracheal dose of 0.2 mg per body weight to male C57BL/6JJmsSlc mice (100 μL/body). A transient decrease in body weight was observed in the test substance administration group 3 days after dosing. Hematological tests revealed a significantly increased eosinophil count and a significantly decreased reticulocyte count in the test substance administration group at the 3-day mark, but not at the 28-day mark. However, increased lung weights were noted in the test substance administration group at both 3 and 28 days post-administration.No abnormalities were observed at necropsy; histopathologic examination consistently showed a moderate accumulation of alveolar macrophages in all cases at 3 and 28 days after administration. These changes were interpreted as adaptive responses to the intratracheal administration of the test compound.Inhalation of a single intratracheal dose of fib-CNF into the lungs showed that CNF-induced inflammation persisted even after 28 days. However, fib-CNF-induced lung inflammation was reduced over time.Inhalation of a single intratracheal dose of fib-CNF into the lung showed that CNF-induced inflammation persisted even after 28 days. Further safety evaluation is needed for CNF with different fiber lengths, widths, and surface properties.

### 5.2. Oral Administration of Fib-CNF to Mice

5.To evaluate the effects of repeated administration, fib-CNF was administered orally by gavage to male C57BL/6JJmsSlc mice (10 mice/group) at doses of 0 (vehicle: water for injection) and 400 mg/kg/day for 28 days.6.Body weight, necropsy, and histopathologic examination of the gastrointestinal tract did not reveal any changes indicative of effects of test substance administration. That is, no overt toxic changes were observed because of the administration of the test substance via fib-CNF.

## Figures and Tables

**Figure 1 toxics-12-00121-f001:**
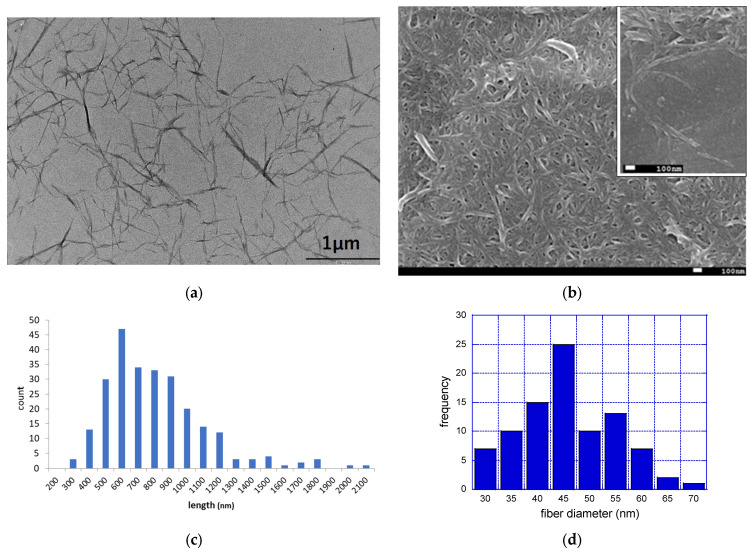
TEM and FE-SEM images of fib-CNF (BiNfi-s FMa-10002) nanofibers and fiber characteristics (length and diameter). (**a**) TEM image of fib-CNF suspensions; (**b**) Fe-SEM image of fib-CNF aggregate suspensions. Inlet: different view; (**c**) fiber length distribution; and (**d**) fiber diameter distribution.

**Figure 2 toxics-12-00121-f002:**
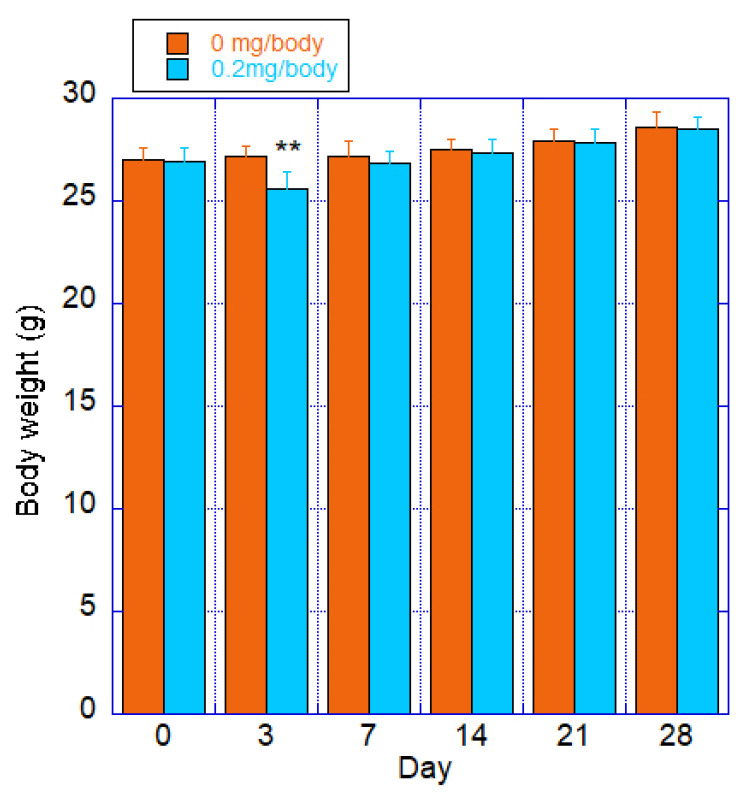
Body weight of mice after a single intratracheal administration of fib-CNF; Orange: control (100 μL of distilled water only); Blue: fib-CNF (0.2 mg/100 μL). ** ANOVA revealed a significant difference between the control and the CNF intratracheal groups, with a *p*-value of 0.03. Additional information can be found in Appendix A.

**Figure 3 toxics-12-00121-f003:**
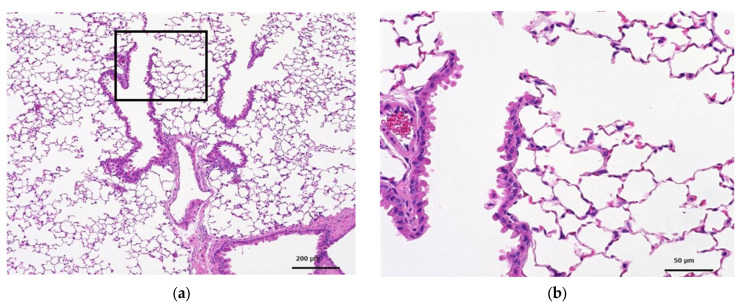
HE-stained lung tissue 3 days after a single intratracheal administration of fib-CNF to mice. (**a**) Control (100 μL distilled water only), (**b**) enlarged image of the area of the square in (**a**), (**c**) fib-CNF (0.2 mg/100 μL), and (**d**) enlarged image of the area of the square in (**c**); CNF aggregates (arrow mark) and macrophages are observed.

**Figure 4 toxics-12-00121-f004:**
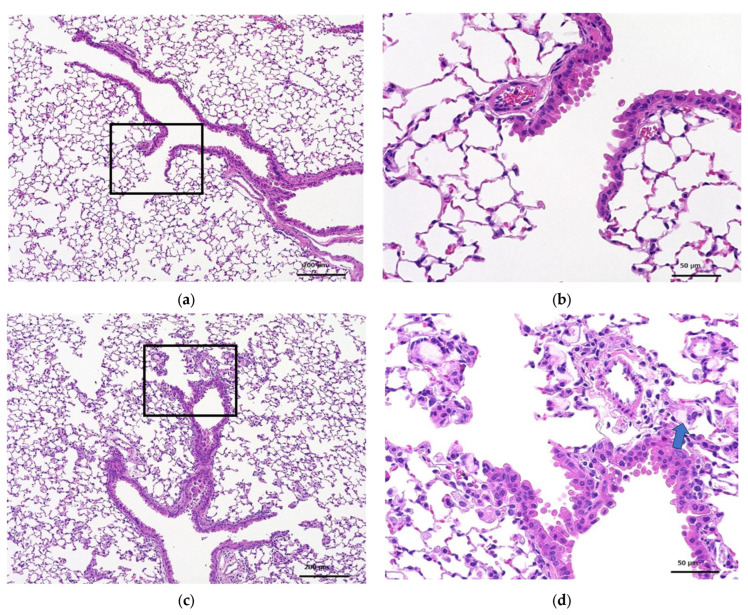
HE-stained lung tissue 28 days after a single intratracheal administration of fib-CNF to mice. (**a**) Control (100 μL distilled water only), (**b**) enlarged image of the area of the square in (**a**), (**c**) fib-CNF (0.2 mg/100 μL), and (**d**) enlarged image of the area of the square in (**c**); CNF aggregates (arrow mark) and macrophages are also observed.

**Figure 5 toxics-12-00121-f005:**
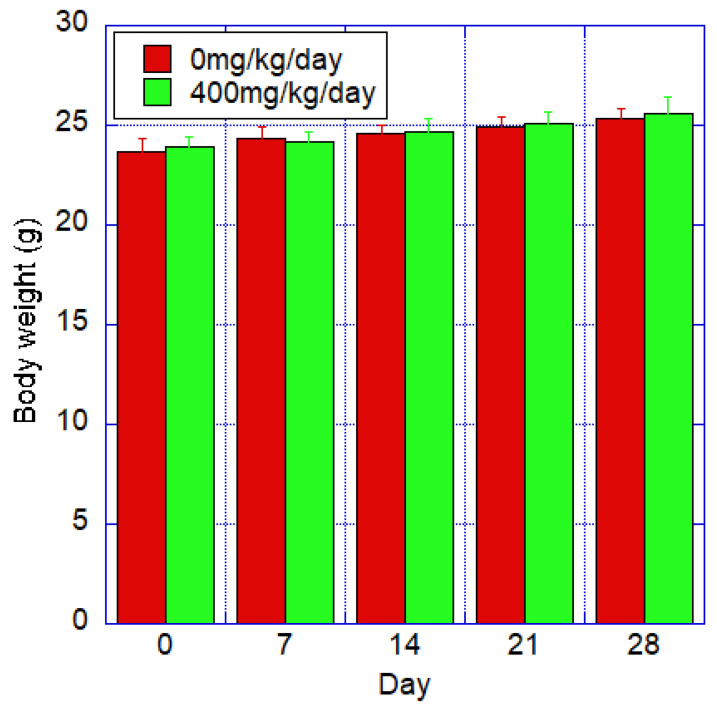
Body weight of mice after oral administration of fib-CNF for 28 days of control and forced administration groups of 400 mg/kg/day of fib-CNF. No significant difference from the control group. Additional information can be found in Appendix A.

**Figure 6 toxics-12-00121-f006:**
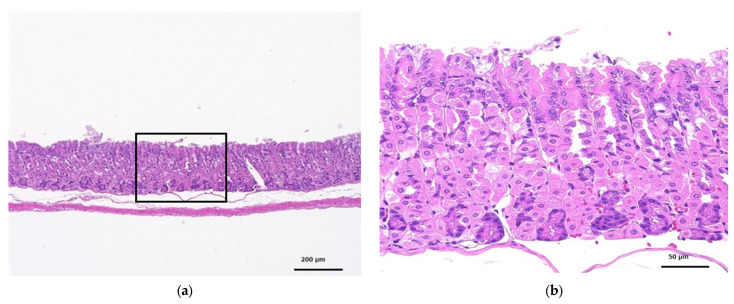
HE-stained histopathology of the glandular stomachs of mice orally administered fib-CNF for 28 days. (**a**) Control, (**b**) enlarged image of the area of the square of (**a**), (**c**) forced administration of fib-CNF 400 mg/kg/day, and (**d**) enlarged image of the area of the square of (**c**).

**Table 1 toxics-12-00121-t001:** Single intratracheal administration group of mice. Necropsy date (3 days after administration): 24 January 2022; necropsy date (28 days after administration): 18 February 2022.

Group	Test Substance	Dose (mg/Body)	Concentration of Administered Solution (wt%)	Dose Volume (μL/Body)	Number of Animals	Necropsy Date
01M	–	0 ^a^	0	100	5	3 days after administration
02M	fib-CNF BiNfi-s FMa-10002	0.2	0.2	100	5	
03M	–	0 ^a^	0	100	5	28 days after administration
04M	fib-CNF BiNfi-s FMa-10002	0.2	0.2	100	5	

^a^: Dose with distilled water only.

**Table 3 toxics-12-00121-t003:** Oral administration group of mice (Autopsy date: 17 February 2022).

Group	Test Substance	Dose (mg/kg/day)	Concentration of Administered Solution (wt%)	Number of Animals
01N	–	0 ^a^	0	10
02N	fib-CNF BiNfi-s FMa-10002	400 ^b^	2	10

^a^: Dose with aqueous distilled water. ^b^: Test substance (2 wt% deionized water solution) administered as is.

**Table 4 toxics-12-00121-t004:** Hematology parameters of each group (mean ± standard deviation) measured 3 days post-administration. Additional information can be found in Appendix A.

Specimen	Sex	Dose (mg/Body)	No. of Animals	RBC (×10^4^/µL)	HGB (g/dL)	HCT (%)	MCV (fL)	MCH (pg)	MCHC (g/dL)
**Control**	M	0	4 ^a^	927 ± 29	13.5 ± 0.3	42.5 ± 1.3	45.9 ± 1.6	14.6 ± 0.2	31.8 ± 0.7
**fib-CNF**	M	0.2	5	924 ± 28	13.6 ± 0.2	42.6 ± 0.3	46.1 ± 1.3	14.7 ± 0.3	31.9 ± 0.5
		**PLT** **(×10^4^/µL)**	**Reticulocytes** **(×10^4^/µL)**	**WBC** **(×10^2^/µL)**	**Leukocytes**				
					**Lymphocytes** **(×10^2^/µL)**	**Neutrophils** **(×10^2^/µL)**	**Eosinophils** **(×10^2^/µL)**	**Basophils** **(×10^2^/µL)**	**Monocytes** **(×10^2^/µL)**
		142.6 ± 15.1	44.4 ± 9.7	43.2 ± 10.6	37.7 ± 9.6	3.2 ± 0.6	1.0 ± 0.2	ND	1.3 ± 0.3
		139.0 ± 45.8	29.6 ± 8.2 *	41.8 ± 14.4	34.7 ± 11.7	3.2 ± 1.2	2.2 ± 1.0 *	ND	1.7 ± 0.9

* Significantly different from the control group at *p* < 0.05. ^a^: One sample omitted from the analysis for coagulation. ND: Not detected.

**Table 5 toxics-12-00121-t005:** Hematology parameters of each group (mean ± standard deviation) measured 28 days post-administration. Additional information can be found in Appendix A.

Specimen	Sex	Dose (mg/Body)	No. of Animals	RBC (×10^4^/µL)	HGB (g/dL)	HCT (%)	MCV (fL)	MCH (pg)	MCHC (g/dL)
**Control**	M	0	5	976 ± 16	14.1 ± 0.4	44.1 ± 1.2	45.2 ± 1.1	14.5 ± 0.3	32.0 ± 0.3
**fib-CNF**	M	0.2	5	974 ± 31	14.2 ± 0.3	44.7 ± 0.9	45.9 ± 0.7	14.6 ± 0.4	31.7 ± 0.5
		**PLT** **(×10^4^/µL)**	**Reticulocytes** **(×10^4^/µL)**	**WBC** **(×10^2^/µL)**	**Leukocytes**				
					**Lymphocytes** **(×10^2^/µL)**	**Neutrophils** **(×10^2^/µL)**	**Eosinophils** **(×10^2^/µL)**	**Basophils** **(×10^2^/µL)**	**Monocytes** **(×10^2^/µL)**
		145.1 ± 15.1	46.7 ± 3.0	69.7 ± 18.5	60.0 ± 16.3	5.9 ± 1.6	1.0 ± 0.4	ND	2.9 ± 0.8
		140.9 ± 27.1	43.1 ± 4.4	80.4 ± 25.0	70.5 ± 21.7	6.0 ± 2.0	1.1 ± 0.5	ND	2.8 ± 0.9

No significant difference from the control group.

**Table 6 toxics-12-00121-t006:** Organ weights—group mean values (mean ± S.D.) at 3 days after administration. Additional information can be found in Appendix A.

Specimen	Sex	Dose (mg/Body)	No. of Animals	Body Weight (g)	Lungs (g)
Control	M	0	5	27.0 ± 1.0	0.138 ± 0.006
Fib-CNF	M	0.2	5	25.7 ± 0.4 ^#^	0.194 ± 0.008 **^##^**

^#, ##^: Significantly different from the control group at *p* < 0.05 (^#^) and *p* < 0.01 (^##^).

**Table 7 toxics-12-00121-t007:** Organ weights—group mean values (mean ± S.D.) at 28 days after administration. Additional information can be found in Appendix A.

Specimen	Sex	Dose (mg/Body)	No. of Animals	Body Weight (g)	Lungs (g)
Control	M	0	5	28.6 ± 0.7	0.141 ± 0.008
Fib-CNF	M	0.2	5	28.5 ± 0.6	0.177 ± 0.013 **^##^**

^##^: Significantly different from the control-2 group at *p* < 0.01.

**Table 8 toxics-12-00121-t008:** Histopathological evaluation of lung/bronchial alveolar macrophage aggregation in male mice, 3 days post-administration. Additional information can be found in Appendix A.

Specimen	Dose (mg/Body)	Number of Animals	Scale				
			None	Minimal	Minor	Moderate	Severe
Control-1	0	5	5	0	0	0	0
fib-CNF	0.2	5	0	0	0	5 **	5

Significantly different from the control-1 group. **: *p* ≤ 0.01 determined via Fisher’s exact test.

**Table 9 toxics-12-00121-t009:** Histopathological evaluation of lung/bronchial alveolar macrophage aggregation in male mice, 28 days post-administration. Additional information can be found in Appendix A.

Specimen	Dose (mg/Body)	Number of Animals	Scale				
			None	Minimal	Minor	Moderate	Severe
Control-1	0	5	5	0	0	0	0
fib-CNF	0.2	5	0	0	0	5 **	0

Significantly different from the control-2 group. **: *p* ≤ 0.01 determined via Fisher’s exact test.

**Table 10 toxics-12-00121-t010:** Hematology—group mean values (mean ± S.D.). Additional information can be found in Appendix A.

Specimen	Sex	Dose (mg/Body)	No. of Animals	RBC (×10^4^/µL)	HGB (g/dL)	HCT (%)	MCV (fL)	MCH (pg)	MCHC (g/dL)
**Control**	M	0	4	978 ± 22	14.5 ± 0.3	43.5 ± 0.7	44.5 ± 0.3	14.8 ± 0.2	33.3 ± 0.3
**fib-CNF**	M	400	5	973 ± 20	14.5 ± 0.3	43.5 ± 0.7	44.7 ± 0.4	14.9 ± 0.2	33.2 ± 0.2
		**PLT** **(×10^4^/µL)**	**Reticulocytes** **(×10^4^/µL)**	**WBC** **(×10^2^/µL)**	**Leukocytes**				
					**Lymphocytes** **(×10^2^/µL)**	**Neutrophils** **(×10^2^/µL)**	**Eosinophils** **(×10^2^/µL)**	**Basophils** **(×10^2^/µL)**	**Monocytes** **(×10^2^/µL)**
		154.1 ± 3.1	48.0 ± 2.0	15.1 ± 5.3	13.7 ± 3.5	2.6 ± 1.1	0 ± 0.1	ND	0.5 ± 0.1
		146.1 ± 5.0 *	47.7 ± 1.4	14.8 ± 4.3	12.5 ± 3.8	1.9 ± 0.5	0 ± 0	ND	0.4 ± 0.2

* Significantly different from the control group at *p* ≤ 0.05 determined via Student’s *t*-test.

## Data Availability

The original data presented in the study are included in the article and its Appendix A.

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
