# Peer review of "Assessing the Safety of Mechanically Fibrillated Cellulose Nanofibers (fib-CNF) via Toxicity Tests on Mice: Single Intratracheal Administration and 28 Days’ Oral Intake"

_toxics, 2024, doi:10.3390/toxics12020121_

Round 1
Reviewer 1 Report (New Reviewer)
Comments and Suggestions for Authors
In the present manuscript Yamashita Yoshihiro et al, assess the long-term safety profile of fibrillated cellulose nanofibers. This is an important topic, given safety concerns related to other fibers.
Overall, the data does not fully support the claims made by the authors, as they interpret their data a bit too far. As noted in detail below, several data sets and experimental design justifications are lacking or incomplete. Several critical issues need to be addressed before considering this manuscript for publication:
- P2 - The introduction is repetitive and disconnected at times. Furthermore, the authors digress for the main topic. This section should be redrafted in a shorter more concise way, providing background on the existing research and clearly stating the problem they want to address.
- P3 (L129) - “In the case of oral administration, fib-CNF was forcefully administered intratracheally” - please correct the mode of administration.
- P3 – Overall, there is a lack of key experimental details to reproduce the assays described in the paper and considerations about what method should be used. On line 142, the authors discuss different methodologies that can be employed for the same assay. Although this is a relevant discussion, it should be moved to the results/discussion section. The methods section should describe exactly what the authors did.
- P3 (L145) – Authors talk about the use of CNF in aqueous solution that they proceed to dilute in pure water. Please describe the content of the first solution.
- P3 (L146) – What do the authors mean by “stability of a CNF aqueous solution in water”?
- P4 (L150) – How did the authors determine the potential impurities of the fib-CNF?
- P4 – In Figure one, there is no information on how the authors obtained the SEM images and how the size was measured.
- P4 (L159) – Why specify that 32 mice were bought if only 20 were used?
- P5 (L167) – Same as observation 3.
- P5 – Table 1: What is a water medium? Is it water alone?
- P5 (L177) – Is it a 2% or a 0.2% solution?
- P6 (L204) – Please clarify what you mean by “youngest animal number”.
- P6 – Table 2: only the name of the methods is stated, but no experimental details are provided.
- P7 (L247) – Same as observation 3.
- P7 – Table 2: Please clarify what is a “2% pure aqueous solution”, as the dose given is 400 mg/kg/day. Moreover, why was this dose chosen? Is this related to the mean intake in a real-life setting?
- P8 – Why use a different number of animals between groups, for the collection of serum?
- P9 – Figure 2. There are no numbers in parentheses. Also, it is not possible to see where the significant difference is, and between what groups (no mark).
- P12 – How was the scoring done in the histopathological examination? Who performed the assessment?
- P14 (L369) – The authors state that the presence of cellulose was confirmed using a dye. These results must be shown.
- P17 – Table 11: The relevance of the information present in this table needs to be clarified.
- P21 – The authors concluded that mechanically fibrillated cellulose is not toxic, but they only assessed one size (500 nm in length). This might be misleading as the fiber size is a key factor determining toxicity. The authors should justify very well why this specific size was chosen.
Comments on the Quality of English Language
The manuscript could benefit from a native English speaker proofreading the final document as there are numerous instances in the text where the meaning is not understood due to the language used.
Author Response
Thank you for your careful peer review.
We have made the additions and corrections as shown in the attachment, and would appreciate your review.

Reviewer 2 Report (New Reviewer)
Comments and Suggestions for Authors
The authors describe a safety assessment of cellulose nanofibers after intratracheal administration in mice. The manuscript's idea is new and the authors provided in vivo proof of their hypothesis. I have some comments that need to be addressed:
1- It was very difficult to read the manuscript since the uploaded version was full of track-changes (modifications, deletions etc....)
2- The conclusion section needs to be summarized, and written in a paragraph format.
3- Please provide a comprehensive title for table 2
4- A materials section needs to be provided.
5- Table 11 is not comprehensible
Comments on the Quality of English Language
minor editing is required
Author Response
Thank you very much for your peer review.
We have made the additions and corrections as shown in the attachment, and would appreciate your review.

Reviewer 3 Report (New Reviewer)
Comments and Suggestions for Authors
Author Response
Thank you very much for your peer review.
We have made the additions and corrections as shown in the attachment, and would appreciate your review.

Round 2
Reviewer 1 Report (New Reviewer)
Comments and Suggestions for Authors
The authors endeavor to answer all the questions and clarify my doubts and concerns.
I just have a few comments/ questions:
1 - Authors explain the rationale behind choosing the intratracheal dose, but they should comment on the real-life exposure expected with this material, to make it more relevant;
2 - Line 289: No date of necropsy is presented in table 3;
3 - Figure 1: If the result is based on the observation of 5 images, why no SD is presented?
4 - Tables 8 and 9 should be revised, as only show the score for moderate aggregation, and fail to include the other potential scores (I do understand that all mice scored the same, but it seems a bit misleading to only show the result for moderate.)
Comments on the Quality of English LanguageThere are still a few typos/sentence structure problems, that need to be addressed. For example, line 397, line 406, line 444, line 516 among others.
Author Response
We appreciate your careful peer review and look forward to your continued support.
We have made the following additions and revisions to the paper and look forward to your continued review.

This manuscript is a resubmission of an earlier submission. The following is a list of the peer review reports and author responses from that submission.
Round 1
Reviewer 1 Report
Comments and Suggestions for Authors
The comments and suggestions are in word file attached.

Some editorial changes are required (Please see comments for authors).
Reviewer 2 Report
Comments and Suggestions for Authors
This study aimed to evaluate the safety of mechanically defibrillated cellulose nanofibers (fib-CNF) by toxicity test in mice using single intratracheal administration and oral intake for 28 days. The result revealed that repeated oral administration of fib-CNF to male C57BL/6JJmsSlc mice at a dose of 400 mg/kg for 28 days did not yield apparent toxic effects attributed to the test substance administration. There was some practical significance for the evaluation of safety of fib-CNF. But the authors should consider the following points in a revision:
1.What is the basis for repeating oral administration of fib-CNF at a dose of 400 mg/kg to male C57BL/6JJmsSlc mice? The author should provide an explanation in the article.
2.The section of “2. Experiments” were too simple, detailed description should be included.
3.Since the author has purchased a commercial fib-CNF, what is the difference between it and the fib-CNF used in reality, and how representative is it?
4.The number of indicators for evaluating safety are too small, and the tables are not standardized.
Comments on the Quality of English LanguageIt is good.